# Realgar Alleviated Neuroinflammation Induced by High Protein and High Calorie Diet in Rats via the Microbiota-Gut-Brain Axis

**DOI:** 10.3390/nu14193958

**Published:** 2022-09-23

**Authors:** Cong Feng, Aihong Li, Chenhui Yin, Siying Wang, Weiyuan Jin, Yi Liu, Taoguang Huo, Hong Jiang

**Affiliations:** 1Department of Health Laboratory Technology, School of Public Health, China Medical University, No. 77 Puhe Road, Shenyang North New Area, Shenyang 110122, China; 2Laboratory of Research in Parkinson’s Disease and Related Disorders, Health Sciences Institute, China Medical University, No. 77 Puhe Road, Shenyang North New Area, Shenyang 110122, China; 3The Key Laboratory of Liaoning Province on Toxic and Biological Effects of Arsenic, No. 77 Puhe Road, Shenyang North New Area, Shenyang 110122, China

**Keywords:** realgar, GHRS, neuroinflammation, microbiota-gut-brain axis

## Abstract

Purpose: Gastrointestinal heat retention syndrome (GHRS) often occurs in adolescents, resulting into nervous system injury. Realgar, an arsenic mineral with neuroprotective effect, has been widely used to treat GHRS. However, its mechanism of action remains unknown. Methods: A GHRS rat model was established using a high protein and high calorie diet. We performed macroscopic characterization by assessing bowel sounds, hot/cold preference, anal temperature, and fecal features. Atomic fluorescence spectroscopy was employed to evaluate brain arsenic level while hippocampal ultrastructural changes were analyzed using transmission electron microscopy. In addition, inflammatory cytokines and BBB breakdown were analyzed by western blotting, immunofluorescence assays, and immunohistochemistry staining. We also evaluated hippocampal metabolites by LC-MS while fecal microorganisms were assessed by 16S rDNA sequencing. Results: Our data showed that the high protein and high calorie diet induced GHRS. The rat model depicted decreased bowel sounds, increased fecal characteristics score, preference for low temperature zone, and increased anal temperature. In addition, there was increase in inflammatory factors IL-6, Iba-1, and NF-κB p65 as well as reduced BBB structural protein Claudin-5 and Occludin. The data also showed appearance of hippocampus metabolites disorder and fecal microbial imbalance. Realgar treatment conferred a neuroprotective effect by inhibiting GHRS-specific characteristics, neuroinflammatory response, BBB impairment, metabolites disorder, and microbial imbalance in the GHRS rat model. Conclusion: Taken together, our analysis demonstrated that realgar confers a neuroprotective effect in GHRS rats through modulation of the microbiota-gut-brain axis.

## 1. Introduction

Gastrointestinal heat retention syndrome (GHRS) is a metabolic syndrome which is associated with increased gastrointestinal heat caused by a high fat and calorie diet [1]. GHRS often results in damage of the nervous system and affects development of adolescents, a developmental stage with high burden of GHRS. Previous studies have associated GHRS with inflammation caused by imbalance of immune factors as well as dry stool due to intestinal flora disorder [2,3,4]. The GHRS symptoms include inflammation, a preference for cold, aphthous ulcers, yellow urine, dry stool, constipation, and redness of the tongue [5]. Intestinal microflora can affect the nervous system and intestinal functions, influence the activities of immune system and the secretion of immune factors. Previous data in a constipation model showed that imbalance of intestinal microbiota was a key regulatory factor in neuroinflammation, which participates in central nerve cell damage through microbiota-gut-brain (MGB) axis, which leads to cognitive dysfunction [6,7,8]. Further studies have shown that the microbiota regulates brain development, stress response, cognitive function, and other central nervous system (CNS) activities via the MGB axis [9,10]. Therefore, MGB axis plays an important role in maintaining gastrointestinal functions and brain health.

Realgar, which contains arsenic, has been used as a traditional Chinese medicine, and is incorporated in Niuhuang Jiedu tablets, Angong Niuhuang pills, and Liu Shen Wan [11]. Niuhuang Jiedu tablets are used for GHRS, and realgar may be playing a vital role in its activities. The arsenic component in realgar can pass through the blood-brain barrier (BBB) and accumulate in the brain [12]. Current studies have mainly focused on the toxic effects of realgar in normal organisms [13,14,15]. However, realgar has been used for treatment of diseases, and confers a therapeutic effect [16,17,18]. Thus, the protective effects of realgar on CNS in pathological states have received a lot of attention. 

In this study, we employed metabolomics and microbiomics analyses to study the neuroprotective mechanisms of realgar in a GHRS rat model and identify its molecular targets. These results provide the theoretical basis and experimental data for re-evaluation of the pharmacology of realgar.

## 2. Materials and Methods

### 2.1. Chemicals and Regents

Realgar was purchased from Shenyang Medicine Company (Shenyang, China) while arsenic standard was obtained from Beijing chemical plant (Beijing, China). Loperamide was acquired from Shanghai zhaohui pharmaceutical (Shanghai, China) while antibodies for matrix metalloproteinase 9 (short for MMP-9, rabbit, ab76003), Occludin (rabbit, ab216327), and ionized calcium binding adaptor molecule-1 (short for Iba-1, rabbit, ab178847) were purchased from Abcam plc (Cambridge, UK). Claudin-5 (rabbit, SAB4502981) was obtained from Sigma-Aldrich Trading Co., Ltd. (Shanghai, China), while interleukin-6 (short for IL-6, rabbit, 21865-1-AP), GAPDH (rabbit, 10494-1-AP), and β-actin (rabbit, 20536-1-AP) were purchased from Proteintech Group, Inc (Wuhan, China). Nuclear factor kappa-B (short for NF-κB, rabbit, 8242) was obtained from Cell Signaling Technology (Danvers, MA, USA).

### 2.2. Generation of the GHRS Rat Model

Specific pathogen-free (SPF) SD female rats (3 weeks of age, 55 ± 5 g of weight) were housed in SPF experimental animal center, China Medical University (experimental animal license number SCXK (Liao) 2015–0001). The environment was maintained at 20–25 °C with a relative humidity of 40–70% and an alternate 12 h light/dark cycle. After a one-week period of acclimation, fifty-one rats were randomly divided into three groups, which included the CON group (common diet), GHRS group (high protein and high calorie diet), and GHRS + REA group (high protein and high calorie diet + 1.8 g/kg realgar). The high protein and high calorie feed contained dried fish: full-fat milk powder: wheat flour: pure soy flour at a ratio of 1:1:1:2. The rats were administered with loperamide from the 46th to 70th day, and realgar was gavaged from the 56th to 70th day. Thereafter, the animals were sacrificed for follow-up tests (Figure 1H).

### 2.3. Macroscopic Characterization of Rats

#### 2.3.1. Bowel Sounds Evaluation

The bowel sounds of the rats were recorded for 5 min by a stethoscope in a quiet room. The test was performed by two investigators in a blinded experimental setup.

#### 2.3.2. Hot/Cold Preference Analysis

A hot/cold pain threshold detector was used to test the hot/cold preference of the rats. In the training phase, both sides of the machine were set at 25 °C, and then a rat was placed into the instrument and allowed to explore the environment for 180 s once daily for three days. In the testing phase, the low-temperature area was set at 15 °C, while the high-temperature area was set at 40 °C. The time spent in the low-temperature area was recorded for 5 min.

#### 2.3.3. Anal Temperature Measurement

A BAT–12 microprobe thermometer was used to test the anal temperature of the rats.

#### 2.3.4. Fecal Characteristics

Fecal characteristics were assessed by the Bristol stool scale in terms of feces hardness from one to seven point. One point corresponded to first level of feces hardness, while seven point was recorded to indicate the last level of feces hardness. The test was performed blindly by two investigators.

### 2.4. Assessment of Arsenic Level in the Brain

Brain tissues were digested with HNO_3_ (37%) at room temperature for 48 h, and then underwent centrifugation before dilution. Thiourea-ascorbic acid (12.5%) was added and left standing at room temperature for 30 min. The level of arsenic in the samples was analyzed using atomic fluorescence spectroscopy.

### 2.5. Analysis of Ultrastructural Changes in the Hippocampus

After being anaesthetized, the rats were perfused with PBS, and then the hippocampus tissues were immersed in 2.5% glutaraldehyde. The tissues were fixed with 1% osmium tetroxide, hydrated by a series of graded ethanol, embedded in resin, and then cut into slices. Thereafter, the slices were stained with 4% uranyl acetate and 5% lead citrate. Ultrastructural changes of neurons, mitochondria, synapses, and myelin were analyzed by transmission electron microscopy.

### 2.6. Western Blotting

Cerebral cortex tissues were homogenized in RIPA lysis buffer and then centrifuged for 25 min at 12,000 rpm at 4 °C. The supernatants were collected to determine the protein concentration. After boiling, equal amounts of denatured total protein were resolved in SDS-PAGE and transferred to a polyvinylidene difluoride membrane. The membranes were blocked with 5% non-fat milk for 2 h at room temperature and then incubated with primary antibodies against IL-6 (1: 500), Iba-1 (1: 1000), NF-κB p65 (1: 1000), Occludin (1: 1000), Claudin-5 (1: 500), MMP-9 (1: 1000), along with GAPDH (1: 5000) and β-actin (1: 1000) at 4 °C overnight. The membranes were washed four times in TBST and then incubated with horseradish peroxidase-conjugated secondary antibodies (1: 1000) for 1 h at room temperature. Thereafter, the blots were washed four times in TBST and then developed using electrochemiluminescence reagents. The densities of the protein bands were quantified using Image J software (version 1.8.0, https://imagej.en.softonic.com/, accessed on 3 August 2022).

### 2.7. Immunofluorescence Assay

Hippocampal tissues were cut into 5 μm thick paraffin sections. The hippocampus sections were deparaffinized in xylene, hydrated with a series of graded alcohol, and then heated in a microwave oven for antigen retrieval. The sections were then blocked with 5% BSA, followed by overnight incubation with NF-κB p65 (1:200) and Iba-1 (1:100) primary antibodies at 4 °C. After washing three times in PBS, the sections were incubated with fluorescent secondary antibodies for 1 h, washed in PBS, and then counterstained with DAPI. The sections were analyzed under a fluorescence microscope.

### 2.8. Immunohistochemical Staining

The hippocampus sections were incubated with primary antibodies against IL-6 (1:100), Claudin-5 (1:200), Occludin (1:100), and MMP-9 (1:100) overnight at 4 °C. Thereafter, the slices were incubated with specific IHC detection reagent for 2 h at 37 °C, then observed under a light microscope.

### 2.9. Detection and Analysis of Hippocampal Metabolites by Liquid Chromatography-Tandem Mass Spectrometry (LC-MS)

Methanol (300 μL) and 20 μL of an internal standard were added to 100 μL of the hippocampal homogenate. The mixture was vortexed for 30 s and then sonicated in an ice-water bath for 5 min. The mixture was incubated at room temperature for 120 min, and centrifuged at 13,000× *g* for 15 min at 4 °C. Thereafter, the supernatant was transferred into a 2 mL sample vial for LC-MS analysis.

### 2.10. Detection and Analysis of Fecal Microorganisms by 16S rDNA Sequencing Technology

Fresh fecal samples were frozen at −80 °C immediately when sacrificing the established model. Microbial DNA was extracted using HiPure stool DNA kit and then underwent PCR amplification followed by sequencing. The V3–V4 region of the 16S rDNA gene was amplified by PCR using specific primers (341F: CCTACGGGNGGCWGCAG, and 806R: GGACTACHVGGGTATCTAAT) which had a barcode with an eight-base sequence unique to each sample. Effective tags were obtained after removal of chimeric sequences. Sequences were clustered into operational taxonomic units (OTUs) at a 97% sequence similarity level using UPARSE (version 9.2.64) pipeline. The tag sequence with the highest abundance was selected as a representative sequence within each cluster. Venn analysis was performed in R project VennDiagram package (version 1.6.16, Bell Labs Technology Showcase, Murray Hill, NJ, USA) in different groups. Alpha index comparison was computed by Tukey’s HSD test and Kruskal–Wallis H test in R project Vegan package (version 2.5.3, Bell Labs Technology Showcase, Murray Hill, NJ, USA). Shannon, Chao1, and ACE indexes were calculated in QIIME (version 1.9.1, University of Colorado, Denver, CO, USA). Weighted UniFrac distances were generated with principal coordinates analysis (PCoA) and were plotted in R project Vegan package (version 2.5.3, Bell Labs Technology Showcase, Murray Hill, NJ, USA). The abundance statistics of each taxonomy were visualized in all levels using Krona (version 2.6, http://krona.sourceforge.net, accessed on 3 August 2022). The relative abundance at different taxonomic levels was calculated separately to find the indicator species. Biomarker features between groups were searched by LEfSe software (version 1.0, http://huttenhower.sph.harvard.edu/lefse/, accessed on 3 August 2022) and pROC package (version 1.10.0 http://expasy.org/tools/pROC/, accessed on 3 August 2022). Species abundance was plotted using R project Vegan package (version 2.5.3, Bell Labs Technology Showcase, Murray Hill, NJ, USA) in all levels. 

### 2.11. Statistical Analysis

All the data were represented as mean ± standard deviation and one-way analysis of variance (ANOVA) was applied after homogeneity test of variance. The difference between groups was analyzed using the Tukey’s test. The data was analyzed in GraphPad Prism 5, and *p* < 0.05 was considered statistically significant. Pearson correlation coefficient between inflammatory indicators, metabolic markers, and microbial markers was calculated in R project psych package (version 1.8.4, Bell Labs Technology Showcase, Murray Hill, NJ, USA).

## 3. Results

### 3.1. Effect of Realgar on Macroscopic Characteristics of GHRS Rats

Here, we employed atomic fluorescence spectroscopy to assess the levels of arsenic content in the brain. The data showed that the brain arsenic in GHRS + REA rats was markedly higher compared to brain arsenic in other groups (*p* < 0.05) (Figure 1G), showing that there was accumulation of arsenic in the brain through BBB. We also analyzed macroscopic characteristics in the rats (Figure 1A–F). Compared with the control group, there was decrease in the water consumption and food intake in the GHRS group and in the GHRS + REA group (*p* > 0.05). Whereas there was suppression in the number of bowel sounds, the fecal characteristics score, the residence time in the low temperature zone, and the anal temperature were increased in the GHRS rats compared with the control group (*p* < 0.05). As expected, treatment with realgar successfully rescued the changes (*p* < 0.05). These findings showed that realgar alleviates the GHRS symptoms caused by high protein and high calorie diet.

### 3.2. Effect of Realgar on the Ultrastructure of Hippocampus

The ultrastructure of hippocampal neurons was examined by electron microscopy (Figure 2). The data showed abnormal morphology of neurons and interrupted nuclear membrane, mitochondrial swelling and vacuolation, relatively mild demyelination, and postsynaptic membrane thickening in the GHRS group. These features were alleviated in the GHRS + REA group. These results showed that realgar has a protective role in the hippocampal ultrastructure in the GHRS rats.

### 3.3. Effect of Realgar on Neuroinflammation in GHRS Rats

The expression of inflammatory cytokines was determined by western blotting, immunofluorescence, and immunohistochemistry tests. Results shown in Figure 3 indicate that the protein levels of IL-6 (*p* > 0.05) and Iba-1 (*p* < 0.05) were upregulated in the hippocampus of rats in the GHRS group. This increase was suppressed following realgar treatment, although not significantly (*p* > 0.05). Furthermore, analysis of the nuclear localization of NF-κB p65 in the hippocampus was increased and recovered after realgar treatment. In the cortex, the protein levels of IL-6, Iba-1, and NF-κB p65 were higher in the GHRS group compared with the control group (*p* < 0.05). Treatment with realgar decreased the levels of the aforementioned proteins in GHRS rats (*p* < 0.05). These results suggested that realgar suppressed neuroinflammation in GHRS rats. 

### 3.4. Effect of Realgar on the Integrity of BBB

Immunohistochemistry and western blotting (Figure 4) were conducted to determine the levels of tight junction proteins in the brain of rats. In the hippocampus, MMP-9 protein expression was significantly higher, whereas Occludin protein expression was lower in GHRS rats compared with the control group. However, these changes were obviously antagonized in GHRS + REA rats (*p* < 0.05). Although the protein level of Claudin-5 was lower in GHRS rats, it was not significantly different from the level in the control group (*p* > 0.05). The protein levels of MMP-9, Occludin, and Claudin-5 in the cortex were similar to those in the hippocampus (*p* < 0.05). These results indicated that realgar treatment protected the integrity of BBB in GHRS rats.

### 3.5. Effect of Realgar on the Hippocampus Metabolites

The hippocampal metabolites were analyzed by LC-MS. Results of the orthogonal partial least-square discriminant analysis (OPLS-DA) revealed that different groups were separated with a satisfactory goodness of fit, indicating that the models were robust (Figure 5A,B). A clear metabolite separation was observed between the CON group and GHRS group as well as between the GHRS group and GHRS + REA group.

Subsequently, cluster analysis with VIP > 1 and *p* < 0.05 was performed for metabolites with significant differences. A total of 31 variational metabolites were found between CON group and GHRS group, 22 were found between GHRS group and GHRS + REA group (Figure 5C,D). Next, heatmaps were constructed for the differential metabolites to visualize and depict the correlations among metabolites and inflammation indicators (Figure 6A,B). Results indicated that Iba-1 level was positively correlated with phosphatidylinositol 18:0–20:4 (r = 0.91), heterophylliin E (r = 0.71), tritriacontyl octacosanoate (r = 0.70), and alpha-amanitin (r = 0.70). However, it was negatively correlated with choline (r = −0.76), phosphoric acid (r = −0.62), and hypoxanthine (r = −0.61). IL-6 was positively correlated with 2,4′-dichlorobiphenyl (r = 0.68) and gentisyl-CoA (r = 0.62); but negatively correlated with N-acetylcysteine (r = −0.82), phosphoric acid (r = −0.81), and choline (r = −0.67). The anal temperature was positively correlated with 2,4′-dichlorobiphenyl (r = 0.71) and tritriacontyl octacosanoate (r = 0.60); but negatively correlated with allopurinol (r = −0.76), phosphatidylethanolamine 16:0–22:4 (r = −0.73), choline (r = −0.71), phosphoric acid (−0.64), and N-acetylcysteine (r = −0.62).

The predictive receiver operating characteristic (ROC) curves were generated using the variational metabolites. Analysis of the curves showed that low levels of allopurinol, N-acetylcysteine, and choline may serve as metabolic markers to assess the pro-inflammatory effect of high fat diet in GHRS rats (diagnostic performance = 92%) (Figure 6C). In contrast, high levels of choline and cystathionine may serve as metabolic markers to assess the anti-inflammatory effect of realgar in GHRS + REA rats (diagnostic performance = 77%) (Figure 6D).

The potential biomarkers between CON group and GHRS group were: NBD-stearoyl-2-arachidonoyl-sn-glycerol, xanthine, allopurinol, hypoxanthine, Glu-Gln, adenine, phosphatidylethanolamine 16:0–22:4, L-aspartate, N-acetylcysteine, phosphatidylinositol 18:0–20:4, 5-hydroxydantrolene, carboxylic acid, acetyl-CoA, 3-hydroxypropanoyl-CoA(4-), 2-furoyl-CoA, oxidanesulfonic acid, S-(5-Hydroxy-2-furoyl)-CoA, UDP-N-acetylmuramoyl-L-alanyl-D-glutamate, tritriacontyl octacosanoate, 2-fluorobenzoyl-CoA, 3-methylglutaconyl-CoA, gentisyl-CoA, 3,5-dihydroxyphenylacetyl-CoA, alpha-amanitin, glutathione, heterophylliin E, choline, phosphoric acid, histidine, 2,4′-dichlorobiphenyl, and O-phosphorylhydroxylamine.

The potential biomarkers between GHRS group and GHRS + REA group were: ascorbate, histidine, trichloroacetic acid, L-aspartate, glutamine, phosphatidylinositol 18:1–20:4, carboxylic acid, oxidanesulfonic acid, UDP-N-acetylmuramoyl-L-alanyl-D-glutamate, tritriacontyl octacosanoate, 2-fluorobenzoyl-CoA, glutathione, choline, phosphoric acid, cystathionine, cyclopropanecarboxylate, 3,4-dehydrothiomorpholine-3-carboxylate, 2-chloroethanol, (–)-ureidoglycolate, meso-tartaric acid, glycerone sulfate, and alpha-CEHC-glucuronide.

Subsequently, the signaling pathways associated with the biomarkers were determined using the KEGG pathway database. For CON and GHRS groups, the pathways altered included amino acid metabolism, lipid metabolism, nucleotide metabolism, biosynthesis of other secondary metabolites, metabolism of cofactors and vitamins, metabolism of terpenoids and polyketides, energy metabolism, carbohydrate metabolism, translation, membrane transportation, signaling molecules and interaction, signal transduction, digestive system, nervous system, endocrine system, cancers, neurodegenerative diseases, and endocrine and metabolic diseases, among others. The main metabolic pathways altered for GHRS and GHRS + REA groups were: amino acid metabolism, carbohydrate metabolism, energy metabolism, xenobiotics biodegradation and metabolism, nucleotide metabolism, lipid metabolism, metabolism of cofactors and vitamins, biosynthesis of other secondary metabolites, gene transcription, membrane transportation, signaling molecules and interaction, signal transduction, digestive system, nervous system, endocrine system, cancers, and neurodegenerative diseases, among others.

### 3.6. Effect of Realgar on the Fecal Microorganisms

The effective tags were clustered by 16S rDNA sequencing, and a total of 35,906 OTUs were obtained. A Wayne diagram was constructed for the OTUs depending on their abundance information. Analysis of the diagram revealed significant differences in the structure and quantity of OTUs in different groups (Appendix A).

Notably, the most abundant phyla in the groups were *Bacteroidetes*, *Firmicutes*, *Proteobacteria*, and *Verrucomicrobia*. As shown in Figure 7A, the ratio of *Bacteroidetes* was increased by 10.32%, the ratio of *Firmicutes* was decreased by 15.62%, the ratio of *Proteobacteria* was increased by 4%, and the ratio of *Verrucomicrobia* was increased by 0.44% in the GHRS group compared with the CON group. Moreover, the ratio of *Bacteroidetes* was increased by 0.88%, the ratio of *Firmicutes* was decreased by 0.13%, the ratio of *Proteobacteria* was increased by 1.1%, and the ratio of *Verrucomicrobia* was decreased by 0.83% in the GHRS + REA group compared with the GHRS group. The ratio of *Firmicutes*/*Bacteroidetes* was decreased. These results indicated that the effects of high protein diet on the structure of fecal flora in GHRS rats was significant at the phylum level. However, no significant alterations in *Firmicutes*/*Bacteroidetes* were detected in the GHRS + REA group, indicating that the effect of realgar on the structure of fecal flora in GHRS rats was weak. In addition, realgar treatment only reduced the ratio of *Verrucomicrobia* in GHRS rats.

The α-diversity of microbial communities was explored using the Shannon index, Chao1 index, and ACE index. Results showed that these indices tended to decrease in GHRS rats, and treatment with realgar did not restore the levels of indices (Figure 7B–D). Furthermore, the PCoA revealed that the β-diversity of flora in each group was obviously aggregated, samples in the GHRS group and the CON group were significantly separated, and samples in the GHRS + REA group were distributed closely to the GHRS group, but tended to the CON group (Figure 7E).

In further tests, differential analysis of fecal flora was performed using the LEfSe method (linear discriminant analysis, LDA Score > 2). A total of 80 bacterial clades were found to be significantly different between the CON group and the GHRS group, and 17 were different between the GHRS group and the GHRS + REA group (Figure 8A,B). Next, the flora were compared and analyzed to identify the affected flora by both high calorie feed and realgar. The abundance of *Rhodospirillaceae* (belonging to *Alphaproteobacteria*), *Sellimonas* (belonging to *Lachnospiraceae*), *Anaerostipes* (belonging to *Lachnospiraceae*), *Lachnoclostridium* (belonging to *Lachnospiraceae*), *Gastranaerophilales*, and *Bacteroidaceae* was increased in the GHRS group, but this effect was reversed following realgar treatment. The abundance of *Rikenellaceae* and *Blautia* (belonging to *Lachnospiraceae*) was reduced in the GHRS group, but realgar treatment increased their abundance.

The area under the ROC curve (AUC) > 0.7 was used as the screening standard for ROC curves. As shown in Figure 8C,D, the *Alphaproteobacteria* (AUC = 1.0), *Anaerostipes* (AUC = 1.0), *Bacteroides* (AUC = 1.0), *Gastranaerophilales* (AUC = 1.0), *Lachnospiraceae* (AUC = 1.0), *Rikenellaceae* (AUC = 1.0), *Sellimonas* (AUC = 1.0), and *Blautia* (AUC = 0.77) were found to be early microbial biomarkers for the changes in fecal flora in GHRS rats. In addition, *Sellimonas* (AUC = 1.0), *Gastranaerophilales* (AUC = 1.0), *Lachnospiraceae* (AUC = 0.92), *Blautia* (AUC = 0.76), *Rikenellaceae* (AUC = 0.73), and *Bacteroides* (AUC = 0.725) were found to be early microbial biomarkers for the effect of realgar on the fecal flora in GHRS rats. Comparison of common biomarkers in different groups revealed that high calorie feed increased abundance of *Gastranaerophilales* and *Sellimonas* (belonging to *Lachnospiraceae*), but decreased abundance of *Rikenellaceae* and *Blautia*. These changes were reversed by realgar treatment. Collectively, these results indicated that realgar improved intestinal microbial homeostasis in GHRS rats fed a high calorie diet. 

### 3.7. Correlations among Inflammatory Factors, Metabolic Markers, and Microbial Markers

The associations among cytokines, bacteria, and metabolites in GHRS rats treated with realgar were determined by the Spearman correlation analysis and analysis of heatmaps. Notably, Iba-1 and IL-6 levels were negatively correlated with choline, allopurinol, and N-acetylcysteine; whereas anal temperature was positively correlated with *Gastranaerophilales* in CON and GHRS rats (Figure 9A). Further analysis revealed that Iba-1, IL-6, and anal temperature were negatively correlated with choline and cystathionine; whereas IL-6 was positively correlated with *Rikebellaceae* in GHRS rats treated with realgar (Figure 9B). These results showed that inflammatory indexes were strongly correlated with metabolic markers and microbial markers, and realgar treatment conferred neuroprotection on GHRS rats via the MGB axis.

## 4. Discussion

GHRS is defined by increased gastrointestinal heat with clinical manifestations including fever, sweating, halitosis, thirst and preference for cool drink, dry stool, yellow urine, and tongue redness [3]. GHRS is induced by intake of a high-calorie and high-fat diet, which mostly occur in children and adolescents, and in more females than males [1]. Previous clinical studies indicated that core symptoms of GHRS in children were mainly digestive symptoms, inflammatory reactions, and nervous system symptoms, such as dry stool, poor appetite, vomiting, recurrent respiratory tract infections, night restlessness, and irritability [1,5]. Realgar, an arsenic tetrasulfide compound, is a highly recognized traditional Chinese medicinal product that has been widely used to treat various diseases such as inflammatory diseases and nervous system injury [17,19]. In this study, we successfully built a GHRS rat model to explore the neuroprotective mechanisms underlying the effects of realgar in GHRS induced by a high-calorie diet.

MGB axis refers to crosstalk between the brain and gut which involves several overlapping pathways containing neuroendocrine, gut microorganisms, bacterial metabolites and neuromodulatory molecules [20,21]. Previous studies have reported that gut microbiota play a vital role in the synthesis of neurotransmitters and neuromodulators, which influence gut-brain communication and nervous system functions [22,23]. Alterations in the composition of microorganisms due to dietary intake mediate the development of several diseases [24,25,26]. First, the gut microbiota is a critical regulator in priming neuroinflammatory responses to brain injury [27]. It also produces metabolites which affect brain function [28]. In addition, gut microbiota and its metabolites, such as short chain fatty acids (SCFAs) which result from bacterial fermentation of dietary fiber, have an impact on BBB permeability [29]. It has been shown that in a constipation model, intestinal flora imbalance can cause pathological changes in the brain, while the gut microbiota participates in the regulation of brain development, stress response, cognitive function, and other central nervous system activities via the MGB axis [30,31]. In this study, we evaluated the protective effect of realgar on the nervous system in GHRS rats in regard to the MGB axis.

We first investigated macroscopic characteristics, such as fecal shape, bowel sounds, hot and cold tendency, and anal temperature in GHRS rats. Previous data has shown that excessive fat intake is one of the main causes of constipation, and a high-fat diet led to constipation with delayed colon transit time possibly via reduction of colonic mucus in mice [32]. Bowel sounds act as an index of bowel activity, which is a common presenting concern of functional dyspepsia [33]. In addition, the high fat/cafeteria diet could induce brown adipose tissue thermogenesis in an obese rat model [34]. Our findings showed that realgar had therapeutic effects on GHRS rats, rescued the decreased bowel sounds, as well as the addition of fecal characteristics score, residence time in low temperature zone, and anal temperature in the GHRS rat model.

The ultrastructural changes of neurons are often used to estimate nerve injury. Previous studies have shown that the high-fat diet might induce neuronal damage mediated in part by synaptic plasticity alteration and dendritic spine loss [35]. Also, the high-fat diet was shown to induce hippocampal neural cell loss, such as degenerate neurons, damaged mitochondria, and extended cisterns of the endoplasmic reticulum [36]. In our study, we demonstrated that GHRS rats have perturbed nuclear membrane, mitochondrial swelling, demyelination, and synaptic destruction. Realgar had a protective effect on hippocampal neurons of the GHRS rats.

High fat diet can promote the release of inflammatory factors and trigger neuroinflammation [37]. Indeed, the short-term consumption of high-fat diet was related to increased inflammatory signals in the brain, which increased the concentration of TNF-α and IL-1β proteins [38]. Microglia are macrophages with delicate branching processes in the CNS, and their activation is a key indicator of neuroinflammation [39]. In our study, the levels of Iba-1, IL-6, and NF-κB p65 both in the hippocampus and cortex were increased leading to inflammatory responses in the GHRS group. Realgar played an anti-inflammatory role in the GHRS + REA group.

BBB is a critical biological barrier that can protect brain development and maintain its physiological function [40]. High-energy diet consumption suppressed the expression of tight junction proteins, particularly Claudin-5 and Claudin-12 [41]. It was shown that NF-κB signal pathway might activate MMP-9 to degrade tight junction structure of the BBB [42]. In agreement, there was upregulation of the MMP-9 protein, while Claudin-5 and Occludin levels were decreased both in the hippocampus and cortex of the GHRS rats, phenotypes that were restored by realgar in the GHRS + REA group. Thus, GHRS induced NF-κB p65 transcription to synthesize MMP-9, inhibited the protein levels of Occludin and Claudin-5, and increased the BBB permeability, ultimately causing CNS damage. Realgar had a therapeutic effect on the BBB breakdown in the GHRS rats because of the high protein and high calorie diet.

To explore the mechanisms of action of realgar in the GHRS rats, we performed metabolomics analysis of hippocampal metabolites in different groups and analyzed the correlation of the biomarkers and inflammatory indexes. The metabolomics results showed that allopurinol, N-acetylcysteine, and choline are early metabolic markers in GHRS rats, while choline and cystathionine are early metabolic markers in GHRS rats under realgar treatment, which all play a vital role in CNS homeostasis. Previous data demonstrated that allopurinol, a xanthine oxidase enzyme inhibitor, could suppress proinflammatory molecules and oxidative stress in vasculature. Pre-treatment with allopurinol significantly reduced infarct volume, microglia infiltration, astrocyte proliferation and nitrative stress in ischemic brain [43]. Furthermore, N-acetylcysteine had a neuroprotective effect against neurotoxicity by modulating oxidative stress and inflammatory reaction [44]. For example, N-acetylcysteine was shown to reduce the levels of RIPK3, MLKL, NLRP3, IL-18, ASC, iNOS, GFAP, and MMP-9; and downregulate myeloperoxidase activity in cerebral cortex [45]. In addition, choline, an alpha7 nAChR agonist, might be a useful drug in the recovery of brain injury by reducing brain inflammation [46]. Cystathionine modulates amino acid metabolism. For instance, cystathionine β-synthase catalyzes condensation of serine and homocysteine to water and cystathionine, which are then hydrolyzed to cysteine, α-ketobutyrate, and ammonia by cystathionine γ-lyase in the reverse transsulfuration pathway [47].

We then analyzed the effects of high fat diet and realgar on the fecal microbiota in the rats by PCR in V3-V4 region of 16S rDNA. As reported in previous studies, long-term dietary changes lead to major alterations in the composition of gut microbiota [48]. A high-fat diet is directly associated with intestinal microbiota dysbiosis, increased energy-harvesting capacity, and metabolic inflammation [49,50,51]. The Chao1 index, ACE index, and Shannon index showed that the α-diversity of microbial community was suppressed in the GHRS group, which indicated that high protein and high calorie diet alters the structure of fecal flora, which was not fully recovered in the GHRS + REA group. The PCoA results showed obvious separation of the flora in different group; the GHRS group and the CON group were significantly separated. Our data also showed four major bacterial phyla with high abundance, namely *Bacteroidetes*, *Firmicutes*, *Proteobacteria,* and *Verrucomicrobia* in the GHRS rats. In addition, the proportion of *Firmicutes*/*Bacteroides* was significantly decreased. After realgar treatment, the proportion of *Firmicutes*/*Bacteroides* was increased, while *Proteobacteria* was reduced and had a pro-inflammatory effect [52]. The increased abundance of *Alphaproteobacteria* can cause intestinal inflammation and reflects the ecological imbalance or unstable structure of the intestinal microbial community [53]. It was demonstrated that realgar can treat inflammation by reducing the proportion of *Proteobacteria* [54]. In addition, *Firmicutes* played an important role in absorbing energy from the diet and storing fat in intestinal cells, while the imbalance of *Firmicutes/Bacteroides* can lead to metabolic disorders [55]. Furthermore, the increase in *Proteobacteria*, a common feature of dysbiosis of flora, was related to gastrointestinal inflammatory diseases [53]. Therefore, these findings showed that realgar may play a therapeutic role in the GHRS rats by adjusting the abundance and structure of the flora.

Furthermore, microbiomics results showed that *Alphaproteobacteria*, *Anaerostipes*, *Bacteroides*, *Gastranaerophilales*, *Lachnospiraceae*, *Rikenellaceae*, *Sellimonas*, and *Blautia* are early microbial biomarkers defining the changes of fecal flora in the GHRS rats. On the other hand, *Sellimonas*, *Gastranaerophilales*, *Lachnospiraceae*, *Blautia*, *Rikenellaceae*, and *Bacteroides* can be considered as early microbial biomarkers in GHRS rats under realgar treatment. *Gastranaerophilales* belong to the phylum *Cyanobacteria*. It has been shown that the abundance of *Cyanobacteria* in the intestines of colitis mouse model was significantly increased with corresponding inflammatory response, which was suppressed after anti-inflammatory treatment [56]. In the animal model of high-fat diet and obesity, *Lachnospiraceae*, mainly *Blautia*, was significantly decreased, which increased intestinal permeability by reducing the expression of genes encoding tight junction proteins ZO-1 and Occludin [48,49]. More evidence showed that *Blautia* can effectively alleviate the symptoms of intestinal inflammation and decreased barrier permeability associated with high-fat diet and obesity. *Lachnospiraceae* is the main producer of butyrate, which can enhance the integrity of the epithelial barrier and inhibit inflammation. These findings were consistent with the changes demonstrated in our study, which suggested that realgar can inhibit the reproduction of *Cyanobacteria*, promote the growth of *Lachnospiraceae*, especially *Blautia*, to maintain the intestinal microbial homeostasis and reduce intestinal inflammatory reactions in GHRS rats.

MGB axis mediates the development of GHRS through a complicated network. GHRS may be associated with MGB axis imbalance and microorganism effect on brain homeostasis by regulating nerves, immunity, and endocrine [1]. In addition, the microbiota has been shown to influence BBB permeability in mice [40]. Host microbiota also contribute to microglial activation states, especially via SCFAs or aryl hydrocarbon receptor ligands [57]. In addition, microbiota modulate neurobehavior through changing brain insulin sensitivity and metabolism [58]. In this study, we showed that Iba-1 and IL-6 were negatively correlated with choline, Allopurinol, and N-acetylcysteine, while IL-6 and anal temperature were positively correlated with *Gastranaerophilales* in the GHRS rats. In addition, Iba-1, IL-6, and anal temperature were negatively correlated with choline and cystathionine; while IL-6 had a positive association with *Rikebellaceae* in the GHRS rats under realgar treatment. Thus, inflammatory indexes have a strong correlation with both metabolic and microbial markers, which demonstrates that realgar plays a neuroprotective role in GHRS rats via the MGB axis.

## 5. Conclusions

Taken together, our data demonstrate that a high protein and high calorie diet induces GHRS characteristics, neuroinflammatory reaction, BBB impairment, hippocampus metabolites disorder, and fecal microbial imbalance. Realgar restores the steady state of hippocampal metabolite spectrum and fecal microbial spectrum, inhibits inflammatory response, prevents the BBB breakdown, and exerts neuroprotection in GHRS rats through the MGB axis. These data provide a theoretical basis underlying the neuroprotective roles of realgar in the GHRS rats.

## Figures and Tables

**Figure 1 nutrients-14-03958-f001:**
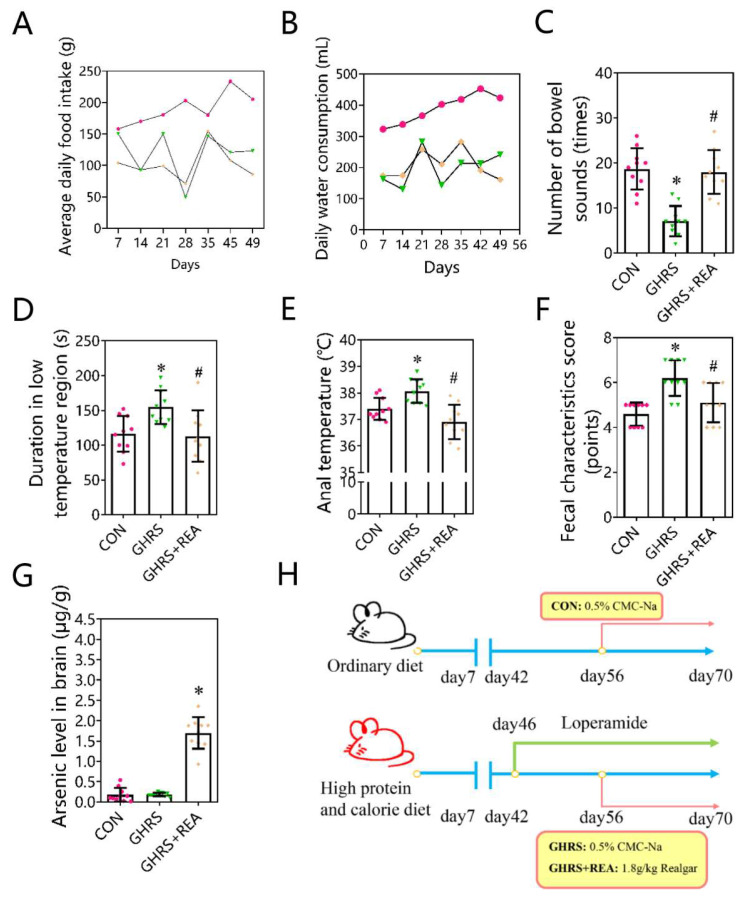
The macroscopic characteristics demonstrating that realgar alleviates the GHRS symptoms in the rats. (**A**) Food intake; (**B**) water consumption; (**C**) number of bowel sounds; (**D**) duration in low temperature region; (**E**) anal temperature; (**F**) fecal characteristics score; (**G**) arsenic level in the brain; (**H**) the experimental procedure of GHRS rat model. The pink spots represent the CON group; green spots represent the GHRS group, and the yellow spots show the GHRS + REA group. The number of animals per group was 10. Each column represents the mean ± SD. * *p* < 0.05 compared with CON group, ^#^
*p* < 0.05 compared with GHRS group.

**Figure 2 nutrients-14-03958-f002:**
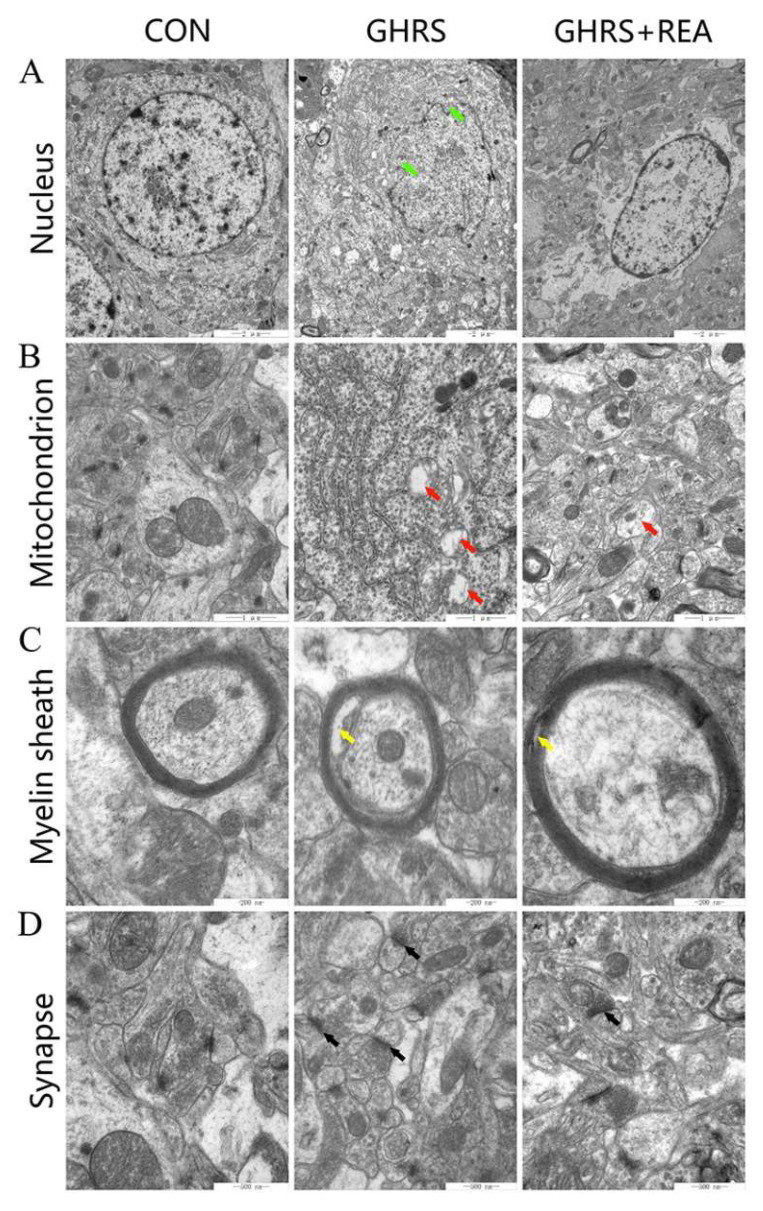
The ultrastructure of hippocampal neurons in rats. (**A**) Results showing the ultrastructure of neuron nucleus; the scale is 2 μm. (**B**) The ultrastructure of the mitochondria; the scale is 1 μm. (**C**) The ultrastructure of myelin sheath; the scale is 200 nm. (**D**) The ultrastructure of the synapse; the scale is 500 nm. The green arrows represent interrupted nuclear membrane, red arrows show mitochondria swelling and vacuolation, yellow arrows show mild demyelination, and black arrows represent postsynaptic membrane thickening.

**Figure 3 nutrients-14-03958-f003:**
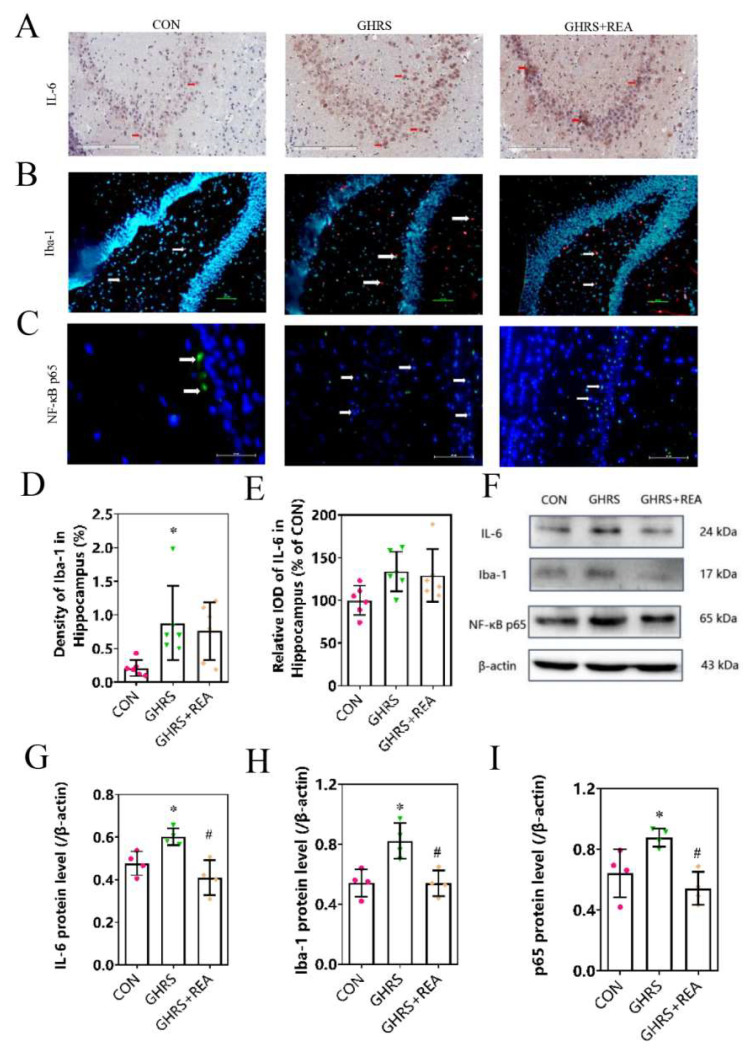
The expression of inflammatory factors in the brain of rats. (**A**) Representative immunohistochemistry photographs showing IL-6 localization in the hippocampus of rats. (**B**) Representative immunofluorescence photographs showing Iba-1 localization in the hippocampus. (**C**) The nuclear translocation of NF-κB p65 in the hippocampus. (**D**,**E**) The protein levels of Iba-1 and IL-6 in the hippocampus, *n* = 6. (**F**) The protein bands of IL-6, Iba-1, and NF-κB p65 in the cortex. (**G**–**I**) The protein levels of IL-6, Iba-1, and NF-κB p65 in the cortex, *n* = 4. Each column presents the mean ± SD. * *p* < 0.05 compared with CON group, ^#^
*p* < 0.05 compared with GHRS group.

**Figure 4 nutrients-14-03958-f004:**
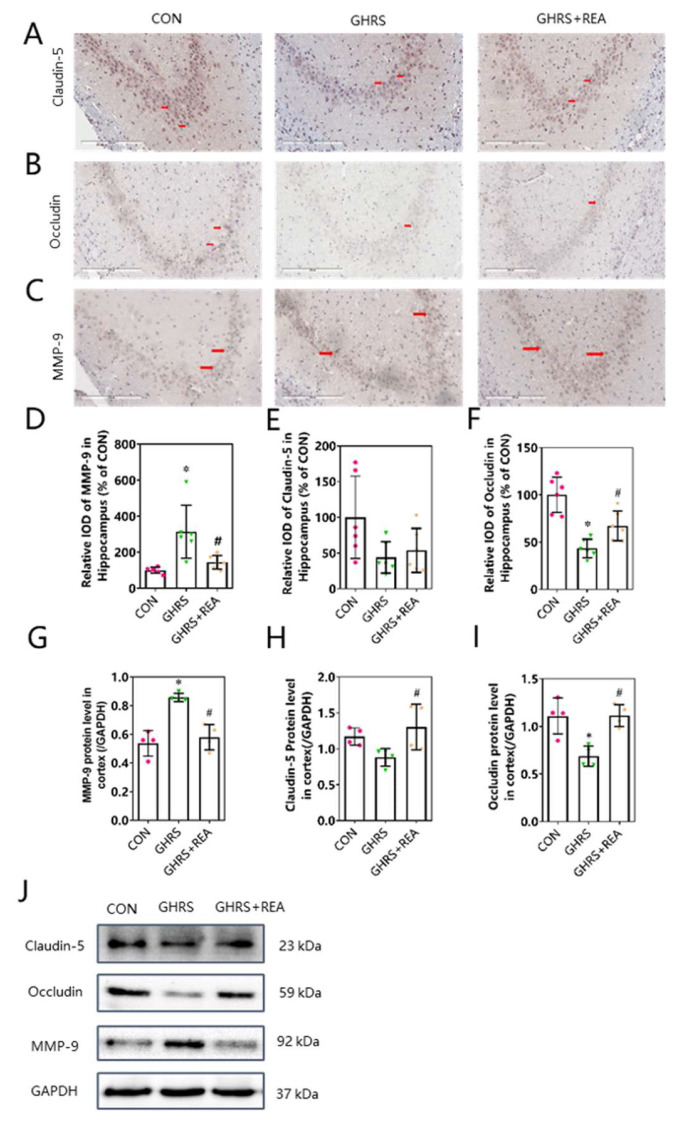
The permeability of BBB in rats. (**A**–**C**) Representative immunohistochemistry photographs showing Claudin-5, Occludin, and MMP-9 expression in the hippocampus of rats. (**D**–**F**) The protein expression of MMP-9, Claudin-5, and Occludin in the hippocampus, *n* = 6. (**G**–**I**) The protein levels of Claudin-5, Occludin, and MMP-9 in the cortex, *n* = 4. (**J**) The protein bands of Claudin-5, Occludin, and MMP-9 in the cortex. Each column shows the mean ± SD. * *p* < 0.05 compared with CON group, ^#^
*p* < 0.05 compared with GHRS group.

**Figure 5 nutrients-14-03958-f005:**
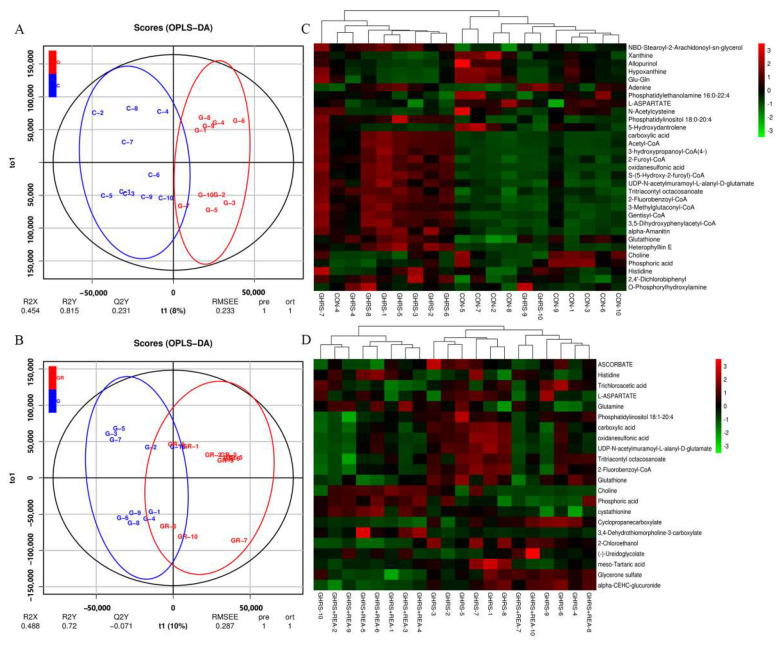
The metabolic markers of each group. (**A**) The OPLS-DA analysis for CON group and GHRS group, *n* = 10. (**B**) The OPLS-DA analysis for GHRS group and GHRS + REA group, *n* = 10. (**C**) The heat map of differential metabolite between CON group and GHRS group, *n* = 10. (**D**) The heat map of differential metabolite between GHRS group and GHRS + REA group, *n* = 10.

**Figure 6 nutrients-14-03958-f006:**
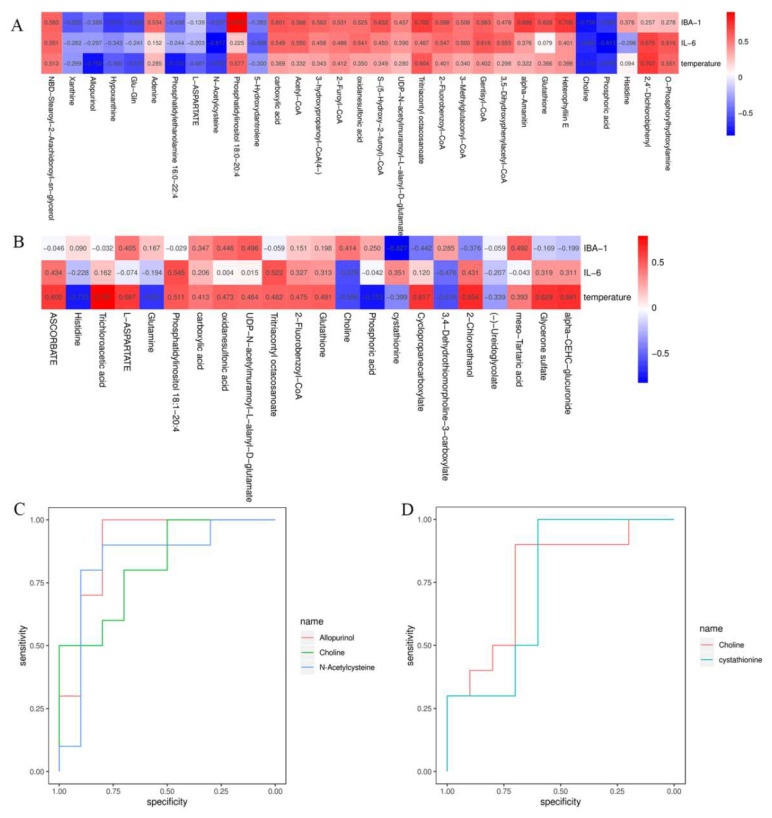
Verification of potential biomarkers. (**A**) The correlations of potential biomarkers between CON group and GHRS group, *n* = 6. (**B**) The correlations of potential biomarkers between GHRS group and GHRS + REA group, *n* = 6. (**C**) The ROC curve of early biomarkers for CON group and GHRS group, *n* = 6. (**D**) The ROC curve of early biomarkers for GHRS group and GHRS + REA group, *n* = 6.

**Figure 7 nutrients-14-03958-f007:**
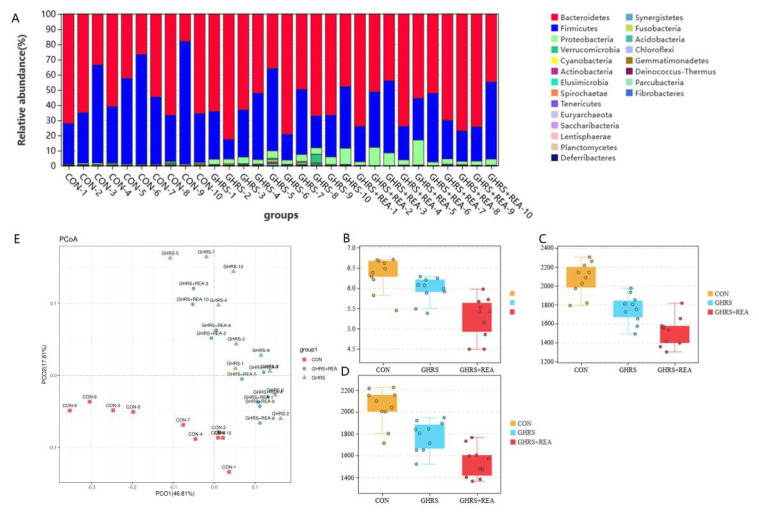
The composition of fecal microorganism in the indicated groups. (**A**) The dominant bacteria in each group. (**B**–**D**) The Shannon index, Chao1 index, and ACE index of microbiota for each group. (**E**) The PCoA analysis of microbiota in the indicated groups.

**Figure 8 nutrients-14-03958-f008:**
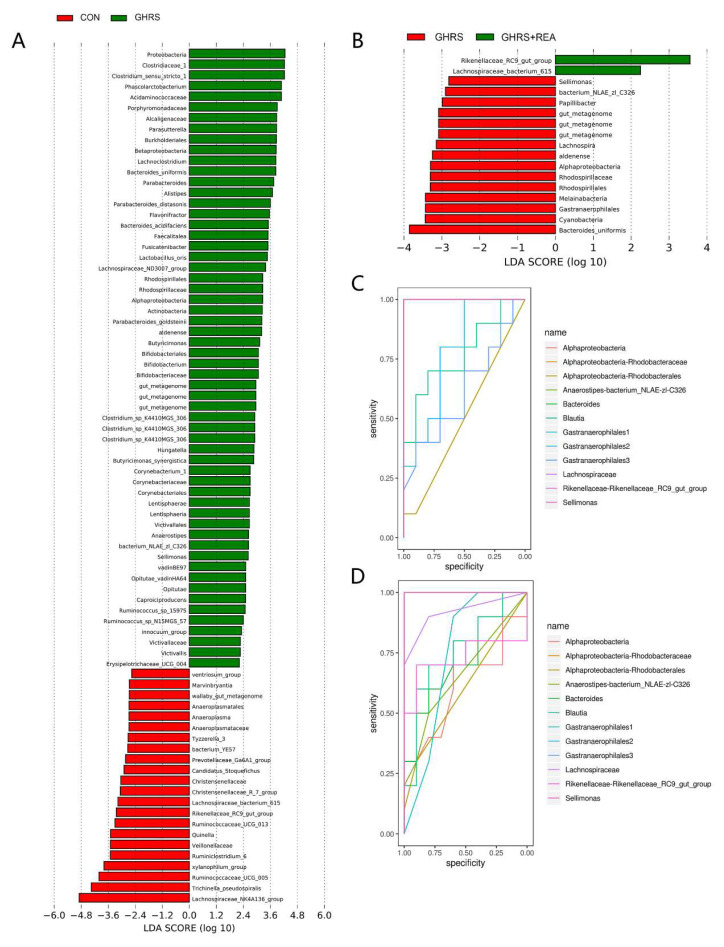
The microbial markers in the indicated groups. (**A**) Histogram showing LDA score for CON group and GHRS group. (**B**) Histogram showing LDA score for GHRS group and GHRS + REA group. (**C**) The ROC curve for CON group and GHRS group. (**D**) The ROC curve for GHRS group and GHRS + REA group.

**Figure 9 nutrients-14-03958-f009:**
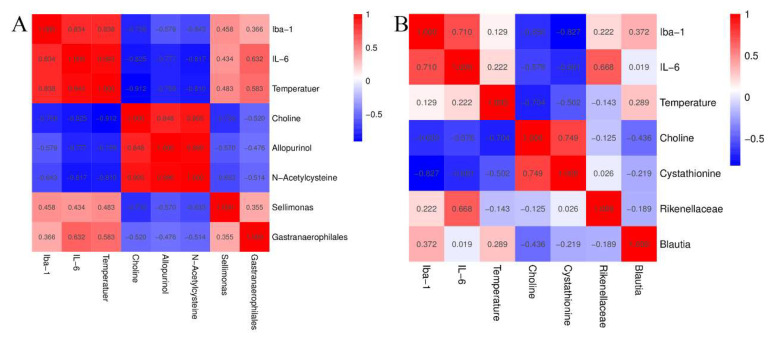
The correlations among inflammatory indicators, metabolic markers, and microbial markers. (**A**) The correlation of indexes between CON group and GHRS group. (**B**) The correlation of indexes between GHRS group and GHRS + REA group.

## Data Availability

Data are available on request from the authors.

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
