# Peer review of "Realgar Alleviated Neuroinflammation Induced by High Protein and High Calorie Diet in Rats via the Microbiota-Gut-Brain Axis"

_nutrients, 2022, doi:10.3390/nu14193958_

Round 1

Reviewer 1 Report

Results are very interesting in order to understand the mechanism of GHRS. 

It would be interesting to understand whether there are safety thresholds for the drug

Reviewer 2 Report

In the paper by Feng et al. metabonomics and microbiomics techniques have been used to study the neuroprotective mechanisms of realgar in a GHRS rat model.

The study is well written and the statistical analyses have been well performed but some clarifications on methods are needed.

Few concerns:

Materials and methods completely lack for the protocol followed to analyse 16S data.

It is unclear if the bioinformatics pipe leads to obtaining an OTU or ASV table.

Please indicate throughout the manuscript if all taxa levels have been used for statistical computing and consequently obtain relative significance. Were the relative abundance tables at different taxonomic levels used separately?

Figure 5 requires complete restyling. Fonts are too small to be read. I suggest splitting the panels into different figures. 

Figure 6 caption states that panels B-D report the Shannon index, Chao1 index, and ACE index were reported but I can see only two boxplots. 
